# Effect of the oil from the fatty tissues of *Crocodylus siamensis* on gut microbiome diversity and metabolism in mice

**Kongphop Parunyakul[1], Aphisara Chuchoiy[1], Sasiporn Kooltueon[1], Phiyaporn Puttagamnerd[1], Krittika Srisuksai[1], Pitchaya Santativongchai [2], Urai Pongchairerk[3], Phitsanu Tulayakul[4], Teerasak E-kobon[5], Wirasak Fungfuang [1]***

**1** Faculty of Science, Department of Zoology, Kasetsart University, Bangkok, Thailand, **2** Faculty of Veterinary Medicine, Bio-Veterinary Sciences (International Program), Kasetsart University, Bangkok, Thailand, **3** Faculty of Veterinary Medicine, Department of Anatomy, Kasetsart University, Bangkok, Thailand, **4** Faculty of Veterinary Medicine, Department of Veterinary Public Health, Kasetsart University, Nakhon Pathom, Thailand, **5** Faculty of Science, Department of Genetics, Kasetsart University, Bangkok, Thailand

* fsciwsf@ku.ac.th

**Data Availability Statement:** All relevant data are within the paper.

## Abstract

Dietary fat can alter host metabolism and gut microbial composition. Crocodile oil (CO) was extracted from the fatty tissues of *Crocodylus siamensis*. CO, rich in monounsaturated- and polyunsaturated fatty acids, has been reported to reduce inflammation, counter toxification, and improve energy metabolism. The aim of this study was to investigate the effect of CO on gut microbiota (GM) in laboratory mice as well as the accompanying metabolic changes in the animals. Forty-five C57BL/6 male mice were randomly divided into five groups and orally administrated either sterile water (control [C]); 1 or 3% (v/w) CO (CO-low [CO-L] and CO-high [CO-H], respectively); or 1 or 3% (v/w) palm oil (PO-low and PO-high, respectively) for 11 weeks. Body weight gain, food intake, energy intake, blood glucose levels, and blood lipid profiles were determined. Samples from colon tissue were collected and the 16S rRNA genes were pyrosequenced to clarify GM analyses. The results showed that there were no differences in body weight and blood glucose levels. Food intake by the mice in the CO-L and CO-H groups was statistically significantly less when compared to that by the animals in the C group. However, neither CO treatment had a statistically significant effect on calorie intake when compared to the controls. The CO-H exhibited a significant increase in serum total cholesterol and low-density lipoprotein but showed a downward trend in triglyceride levels compared to the control. The GM analyses revealed that both CO treatments have no significant influence on bacterial diversity and relative abundance at the phylum level, whereas increases of Choa1 and abundance-based coverage estimator indexes, distinct β-diversity, and Proteobacteria abundance were observed in the PO-high group compared with the C group. Furthermore, the abundance of *Azospirillum thiophilum* and *Romboutsia ilealis* was significantly higher in the CO-L and CO-H groups which could be associated with energy metabolic activity. Thus, CO may be an alternative fat source for preserving host metabolism and gut flora.

**Funding:** This study was supported by Undergraduate Research Matching Fund (URMF), Faculty of Science, Kasetsart University and partially supported by the Faculty of Veterinary Medicine, Kasetsart University, Thailand. The funders had no role in study design, data collection and analysis, decision to publish, or preparation of the manuscript.

**Competing interests:** The authors have declared that no competing interests exist.

**Abbreviations:** ACE, abundance-based coverage estimator; C, control; CO, crocodile oil; CO-H, CO-high; CO-L, CO-low; HDL, high-density lipoprotein; LDA, linear discriminant analysis; LDL, low-density lipoprotein; LEfSe, LDA effect size; MUFA, monounsaturated fatty acid; NMDS, non-metric multidimensional scaling; OUT, operational taxonomic unit; PCA, principal component analysis; PO, palm oil; PO-H, PO-high; PO-L, PO-low; PUFA, polyunsaturated fatty acid; SFA, saturated fatty acid.

## Introduction

The gut microbiota (GM) consists of a complex microorganism community that modulates an intestinal function. GM comprises ~100 trillion organisms, with 35,000 species of bacteria [1]. The high-abundant phyla of GM are Bacteroidetes, Firmicutes, Verrucomicrobia, Actinobacteria, and Proteobacteria, and ~90% of these belong to Bacteroidetes and Firmicutes [2]. The intestinal flora plays important roles in host metabolic regulation by utilizing consumed nutrients. Meanwhile, the host could obtain its fermentation products to play a crucial role in metabolism as a mutualistic relationship [3,4]. Many researchers have reported numerous associations between imbalanced GM (dysbiosis) and metabolic syndromes such as cardiovascular disease, diabetes, and obesity [5–7]. Thus, GM composition and alteration may be a key predictive indicator for the prevention of these metabolic disorders.

Dietary consumption is the first factor that affects the intestinal organisms and has been reported to shape the GM structure [8]. Dietary fat is one of many key modulators to directly influence the host metabolism and GM composition. Edible fatty acids are divided into three main types based on double bonds in the molecules, including saturated fatty acid (SFA), monounsaturated fatty acid (MUFA), and polyunsaturated fatty acid (PUFA). Increasing the ratio of dietary unsaturated fatty acids to SFAs showed improvement of blood lipid profiles and metabolic dysfunction syndromes [9,10]. Previous studies found that a high-fat diet altered the species and amounts of GM, including a decrease of the Bacteroidetes phylum and increase of both Firmicutes and Proteobacteria in the rodent model [11,12]. The Firmicutes/Bacteroidetes ratio was frequently considered as a hallmark of health status [13]. Moreover, the Firmicutes/Bacteroidetes ratio also has been reported to be associated with obesity and metabolic syndrome [14]. Further study demonstrated that Firmicutes involved in energy metabolism by showing a significant number of genes of carbohydrate metabolism [15]. In contrast, a MUFA-rich diet was shown to improve metabolic health by altering the abundance of bacterial taxa [16]. Another study also demonstrated that PUFA supplementation restored the microbiota and inflammatory cell infiltration as well as promoted regulatory T cell recruitment [17]. Thus, different types of dietary fatty acids (SFAs, MUFAs, and PUFAs) are important in modifying the gut microbiome composition, which interplays between the host health status and metabolic disease.

*Crocodylus siamensis*, commonly known as Siamese crocodile, is a freshwater crocodile originally distributed in Southeast Asia. In Thailand, Siamese crocodiles were nourished in the crocodile farm in Nakhon Pathom province for industrial processing and research. Crocodile oil (CO), extracted from abdominal fat tissues from *C. siamensis*, is highly rich in MUFAs and PUFAs compared to other animal fat sources [18]. Previous studies showed that CO exhibited effective improvement of skin rashes, wound healing, and anti-microbial activity [19,20]. Our previous study also demonstrated that CO supplementation could maintain energy metabolism in the liver by preserving hepatic mitochondrial structure and key energy maintaining protein [21]. Moreover, another study found that CO could help in liver detoxification by downregulating the expression of cytochrome P450 1A2 (CYP1A2) in high fat–fed rats [22]. A recent study reported that CO reduced DNA damage and increased cell cycle regulators, which helped to avoid inflammatory disease [23]. However, the impact of CO on GM structure as related to metabolic alterations is largely unknown.

The present study aimed to investigate how CO supplementation modulates the gut microbiome community in mice. We used 16S rRNA gene sequencing to identify specific microbial signatures in the gut of mice fed with CO treatment. We discuss how CO interacts with the GM systems and the relationship between gut microbes and host metabolism. Our research might provide new insights into the potential impact of the use of CO on GM homeostasis, which is associated with host metabolism and intestinal health.

## Material and methods

### CO preparation

Abdominal fat samples were discarded and collected as byproducts from slaughtered *C. siamensis* (3–5 y old) obtained from a crocodile farm in Nakhon Pathom Province, Thailand. CO was extracted by wet cold-pressing method according to Santativongchai et al. [18] after the meat was trimmed and prepared. The samples were pressed through two layers of filter cloth with distilled water at a proportion of 1:1 (w/v). The solution was then left undisturbed until the mixture separated. The upper clear oil fraction was then collected, evaporated, and stored in a sealed container at room temperature.

### Animal care and experimental design

Forty-five male C57BL/6 mice (3 wk old) were obtained from Nomura Siam International (Bangkok, Thailand). The animals were housed in individual cages at $25 \pm 2$°C on a 12-h light/12-h dark cycle with ad libitum access to standard diet and drinking water. Mice were randomly divided into five groups of nine mice each: mice that received sterile water (C group); mice supplemented with 1% (v/w) CO (CO-L group); mice supplemented with 3% (v/w) CO (CO-H group); mice supplemented with 1% (v/w) PO (PO-L group); or mice supplemented with 3% (v/w) PO (PO-H group). Mice were orally administered sterile water, CO, or PO once daily for 11 wk. The fatty acid profile of experimental oil (CO and PO) is presented in Table 1. The total PUFAs and MUFAs contents were higher and SFAs contents were lower in CO than that of PO. The study conducted adhered to the *Guidelines for the Care and Use of Laboratory Animals*. The Ethics Committee of Kasetsart University Research and Development Institute, Kasetsart University, Thailand, approved this study (approval no. ACKU61-VET-088).

### Measurement of body weight, food intake, energy intake, blood glucose, and blood lipid profiles

Body weight, food intake, and energy intake were measured individually between 11:00 and 11:30 AM. Each mouse's food intake was measured daily by weighing the remaining chow.

**Table 1. Fatty acid composition (g/100 g FA) of crocodile oil and palm oil.**

| Fatty acids | Crocodile oil | Palm oil |
|---|---|---|
| Lauric acid (C12:0) | 0.11 | 0.35 |
| Myristic acid (C14:0) | 0.57 | 1.39 |
| Pentadecylic acid (C15:0) | 0.09 | 0.08 |
| Palmitic acid (C16:0) | 19.9 | 43.4 |
| Heptadecanoic acid (C17:0) | 0.16 | 0.13 |
| Stearic acid (C18:0) | 5.42 | 4.5 |
| Arachidic acid (20:0) | 0.8 | 0.1 |
| Oleic acid (C18:1) | 41.07 | 37.5 |
| Myristoleic acid (C14:1) | 0.1 | 0 |
| Palmitoleic acid (C16:1) | 3.83 | 0.54 |
| Linoleic acid (C18:2) | 21.1 | 10.1 |
| Linolenic acid (C18:3) | 0.96 | 0.4 |
| **Total SFAs** | **27.1** | **50.5** |
| **Total MUFAs** | **45.5** | **37.5** |
| **Total PUFAs** | **23.8** | **10.5** |

FA, fatty acid; MUFA, monounsaturated fatty acid; PUFA, polyunsaturated fatty acid; SFA, saturated fatty acid.

Source: Adapted from Santativongchai et al. [18] and Mancini et al. [24].

Energy intake was calculated from the food intake determination as (food intake per mouse x ME from standard chow) + ME from the treatment (ME is the total energy of the rat diets, which is 3.040 kcal/g in standard chow and 2.4 kcal in CO-L, 4.8 kcal in CO-H, 2.6 kcal in PO-L, and 5.2 kcal in PO-H diets). Body weight was also measured weekly throughout the experiment.

On the last day of experiment, blood glucose level was monitored using a blood glucose meter (Accu-Chek Active) after an 8-h overnight fast. Meanwhile, blood samples were collected and centrifuged at 2200 x g for 15 min at 4˚C, and the serum was stored at –20˚C until further analysis. The serum lipid profiles included triglycerides, cholesterol, HDL, and LDL and were enzymatically determined on a HITACHI-7080 automatic biochemical analyzer (Hitachi, Tokyo, Japan).

## Sample collection and DNA extraction

After 11 wk of the experiment, all animals were euthanized with 60 mg/ml pentobarbital sodium by intraperitoneal injection. Colon tissues were collected and immediately frozen at -80˚C until further analyses. Microbiota genomic DNA was extracted from colon samples using the QIAamp Fast DNA stool mini kit (Qiagen, Hilden, Germany). The quality and purity of the genomic DNAs were determined using a NanoDrop 2000 spectrophotometer (Thermo Scientific, Waltham, MA) and 1.5% agarose gel electrophoresis with ethidium bromide staining and visualization by a UV transilluminator (Bio-Rad Laboratories, Hercules, CA). A PCR reaction was used to preliminarily amplify the 16S rRNA gene fragments (V3-V4 region) using 27F (TAC GGY TAC TTG TTA CGA and AGA GTT TGA TCM TGG TCT AG) primers. The PCR cycle included initial denaturation at 95˚C for 3 min and 35 cycles of denaturation at 94˚C for 30 s, annealing at 57˚C for 30 s, and extension at 72˚C for 1 min, followed by the final extension at 72˚C for 10 min. The PCR products were checked and visualized by 15% agarose gel electrophoresis as described earlier.

## 16S rRNA metagenomic sequencing of colon samples

The 16S rRNA gene is composed of conserved and variable regions. Similar to the previous step, this study examined the V3–V4 region for bacterial identification and diversity analysis using a metagenomic approach. The amplified 16S rRNA gene fragments were purified and subjected to a 250-bp paired-end library preparation before sequencing by the IIIumina Nova-Seq 6000 platform (Macrogen, Seoul, Korea). Raw read data were merged and filtered to produce the clean data, which were analyzed by the VSEARCH pipeline to classify OTUs according to 97% identity of valid tags of all samples [25]. Sequences with the maximum abundance in each OTU were selected as representatives. The naive Bayesian Ribosomal Database Project classifier was used to compare and annotate the representative sequences to their corresponding OTUs at each taxonomic rank using the National Center for Biotechnology Information BLAST and SILVA databases [26].

## Data analysis and statistical methods

Quantitative experimental data are expressed as mean ± SEM. Statistical analysis was performed by one-way ANOVA followed by a Tukey's post hoc test in the R project statistical computing package (R Core team, 2019). A $p$ value $<0.05$ was considered as statistically significant.

α-diversity was applied in analyzing complexity of the sample biodiversity through five indices, including observed species, Chao1, Shannon, Simpson, and ACE. These indices were calculated with QIIME version 1.7.0 and displayed with GraphPad Prism version 9.0.0

(GraphPad Software, San Diego, CA). β-diversity analysis was used to evaluate differences across the samples in term of species complexity. The β-diversity was also calculated by QIIME software. Cluster analysis was preceded by PCA and NMDS methods, which were applied to reduce the dimension of the original variables using the FactoMineR package and ggplot2 package in R software (Version 2.15.3).

The relative abundance of GM composition at the phylum and genus levels was displayed by Sankey diagrams using SankeyMATIC (http://sankeymatic.com/). A nonparametric Wilcoxon rank-sum test was performed to compare bacterial classification ratios among the five treatment groups ($p < 0.05$). The comparisons were visualized as a boxplot by the GraphPad Prism software. Enrichment of differential relative abundance of GM among these groups was investigated by linear discriminant analysis effect size (LEfSe) analysis (http://huttenhower.sph.harvard.edu/lefse). The LEfSe analysis evaluated the biomarkers from the relative abundance information of all assigned bacterial OTUs. Significance difference of the gut microbiota was considered at $p$ value <0.05 and an LDA score >2 [27].

## Results

### Effect of CO on body weight, food intake, energy intake, and blood glucose level

As shown in Fig 1 and Table 2, the effects of CO showed no significant differences in body weight growth, final body weight, and body weight gain. However, CO and PO treatment showed a significant decrease in food consumption when compared with the control group. Food intake in the CO-high (CO-H) and PO-high (PO-H) groups was significantly decreased compared to the CO-low (CO-L) and PO-low (PO-L) groups. Nonetheless, the energy intake parameter from each treatment's diet showed that both PO-L and PO-H groups had significantly more calories than did the other three groups, whereas the calorie intake did not differ among the control (C), CO-L, and CO-H groups. Likewise, there was no difference in fasting blood glucose levels among the groups.

### Effect of CO on serum cholesterol, triglyceride, low-density lipoprotein, and high-density lipoprotein level

Measurement of the serum lipid profiles (total cholesterol, triglyceride, low-density lipoprotein [LDL], and high-density lipoprotein [HDL]) among the experimental groups were used to evaluate the favorable effect of CO supplementation (Fig 2). Total cholesterol levels were significantly increased ($p < 0.05$) in mice that received the CO-H treatment compared to the C, CO-L, and PO-L treatment (Fig 2A). Remarkably, triglyceride levels were significantly decreased in the CO-H group when compared to the PO-L group (Fig 2B). Nevertheless, LDL levels were significantly higher in the CO-H group than in the other groups. We did not find a difference in HDL levels among groups. However, no changes in lipid profiles were observed in the CO-L group when compared to the control group.

### Effect of CO on richness and diversity of the gut microbiome community

The number and identity of the operational taxonomic units (OTUs) were evaluated through a Venn diagram, as shown in Fig 3A. The total number of OTUs presented in the C, CO-L, CO-H, PO-L, and PO-H groups were 870, 1,098, 1,078, 1,237, and 1,200, respectively. The number of shared OTUs in all groups was 430.

Higher numbers of OTUs were shared between the PO-L and PO-H groups (794 OTUs) compared to between the CO-L and CO-H groups (712 OTUs). Comparison between the CO

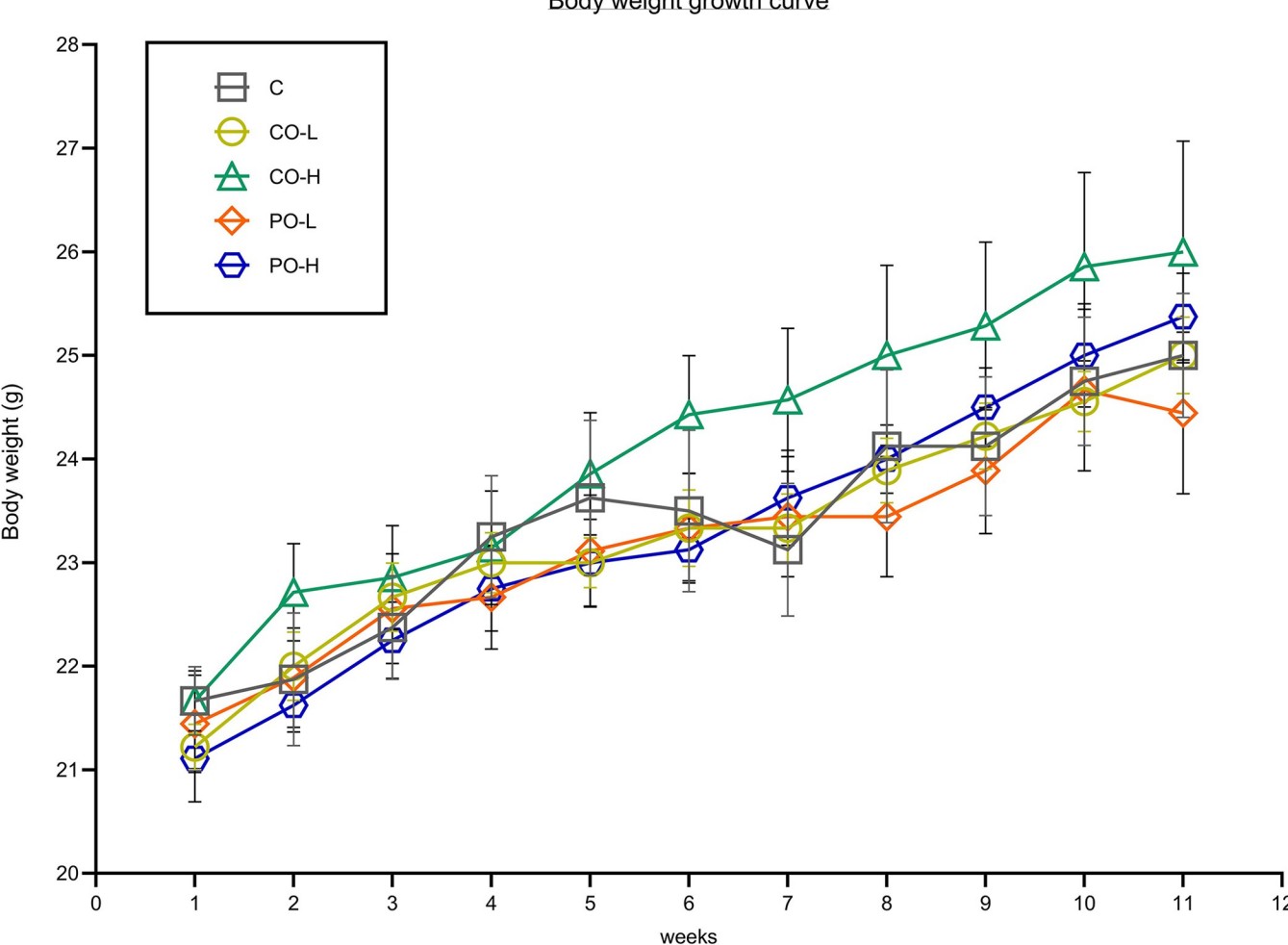

**Fig 1. Effect of CO on body weight growth curve after 11 wk of the experiment.** Data are expressed as mean ± SEM.

and PO groups (725 OTUs between the CO-L and PO-L groups, 740 OTUs between the CO-L and PO-H, 757 OTUs between the CO-H and PO-L groups, and 752 OTUs between the CO-H and PO-H groups) showed a greater number of shared OTUs than for comparison with the

**Table 2. Effect of CO on body weight, food intake, energy intake, and blood glucose levels after 11 wk of the experiment.**

| Parameters | Groups | | | | |
|---|---|---|---|---|---|
| | C | CO-L | CO-H | PO-L | PO-H |
| Initial body weight (g) | 21.67±0.33 | 21.22±0.22 | 21.67±0.29 | 21.44±0.47 | 21.11±0.42 |
| Final body weight (g) | 25.00±0.60 | 25.00±0.37 | 26.00±1.07 | 24.44±0.78 | 25.38±0.42 |
| Body weight gain (g) | 3.25±0.59 | 3.78±0.28 | 4.43±0.90 | 3.00±0.76 | 4.13±0.44 |
| Food intake (g/day) | 2.41±0.05 | 1.73±0.04* | 0.99±0.05*, ** | 1.85±0.04*, # | 1.12±0.03*, ** ## |
| Energy intake (kcal/day) | 7.34±0.15 | 7.58±0.14 | 7.77±0.16 | 8.15±0.14*, ** | 8.55±0.113*, **, # |
| Blood glucose level (mg/dL) | 211.29±10.13 | 223.67±13.01 | 230.14±17.21 | 322.71±41.51 | 311.20±42.71 |

Data are expressed as mean ± SEM.

*, **, #, and ## indicated the comparison within the C, CO-L, CO-H, and PO-L groups at the $p < 0.05$, respectively.

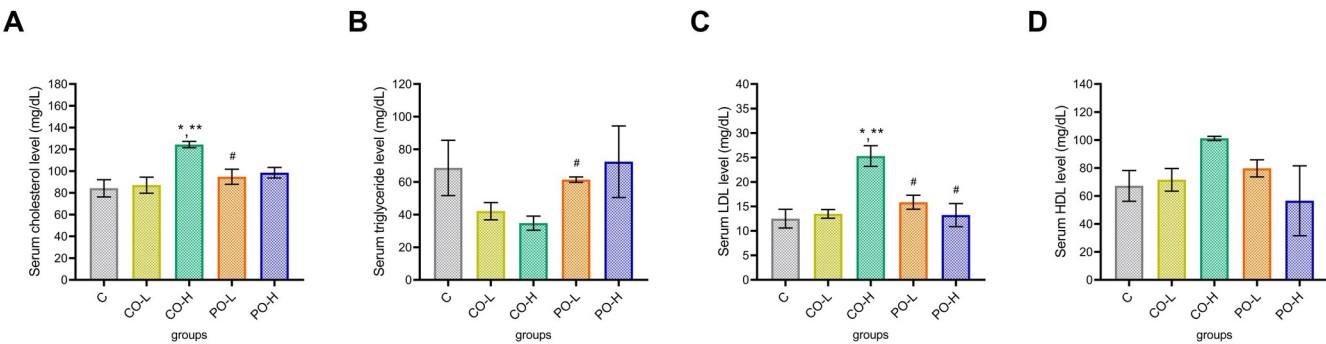

**Fig 2. Effect of CO on serum lipid profiles after 11 wk of the experiment.** (**A**) Cholesterol, (**B**) triglyceride, (**C**) low-density cholesterol, and (**D**) high-density cholesterol levels of mice after 11 wk of the experiment are shown. Data are expressed as mean ±SEM. *, **, and # indicated the comparison within the C, CO-L, and CO-H groups at the $p < 0.05$.

control (631 OTUs between the C and CO-L groups, 612 OTUs between the C and CO-H groups, 617 OTUs between the C and PO-L groups, and 606 OTUs between the C and PO-H groups).

Gut microbiome diversity and richness were investigated using α-diversity represented by Shannon, Simpson, Chao1, abundance-based coverage estimator [ACE], and observed species indexes). As shown in 3B–F, the results showed no differences among the groups based on the Shannon, Simpson, and observed species indexes. Meanwhile, significant differences in the overall microbiome community among the five groups were observed by the Chao1 and ACE indexes (Fig 3D and 3E). The results exhibited that the microbial species richness was significantly increased in the high-dose PO-treated group when compared to the other groups.

The analysis of β-diversity by principal component analysis (PCA) and non-metric multidimensional scaling (NMDS) with stress value = 0.117, as shown in Fig 4, illustrated that the gut microbiota of both PO-treated groups was clearly distinct from the other groups.

## Effect of CO on relative abundance of the gut microbiome community

The relative abundances at the phyla and included genera were displayed using a Sankey diagram (Fig 5A). The five most abundant microbiota phyla across the five experimental groups included Bacteroidetes, Firmicutes, Verrucomicrobiota, Proteobacteria, and Actinobacteriota. The results also revealed that Firmicutes were less abundant in the CO-L and CO-H groups, whereas Verrucomicrobiota was more abundant in the CO-H and PO-L groups (Fig 5B). Further analysis of the genus abundance data showed that the first 10 bacterial genera with the highest relative abundance values were from *Muribaculaceae*, *Akkermansia*, the Lachnospiraceae NK4A136 group, *Lactobacillus*, the *Eubacterium coprostanoligenes* group, the *Eubacterium ruminantium* group, *Alloprevotella*, *Candidatus Saccharimonas*, *Parasutterella*, and *Muribaculum* (Fig 5C).

The linear discriminant analysis (LDA) effect size (LEfSe) analysis at the phylum levels (Fig 5D) showed differences in the abundances of Proteobacteria and Actinobacteriota among the groups. Meanwhile, LEfSe analysis at the genus levels (Fig 5E) showed higher abundances of *Methylophilus* in the C group, *Roseburia* and *Zoogloea* in the PO-L group, and the *Eubacterium coprostanoligenes* group, the *Eubacterium ruminantium* group, and *Comamonas* in the PO-H group, whereas the predominant genus was not observed in the CO-L and CO-H groups. The Proteobacteria phylum had significantly highest abundant in the PO-H group followed by the PO-L, whereas the C, CO-L and CO-H treatment had lower abundance levels (Fig 5F). However, there were no significant differences in relative abundance of the Actinobacteriota

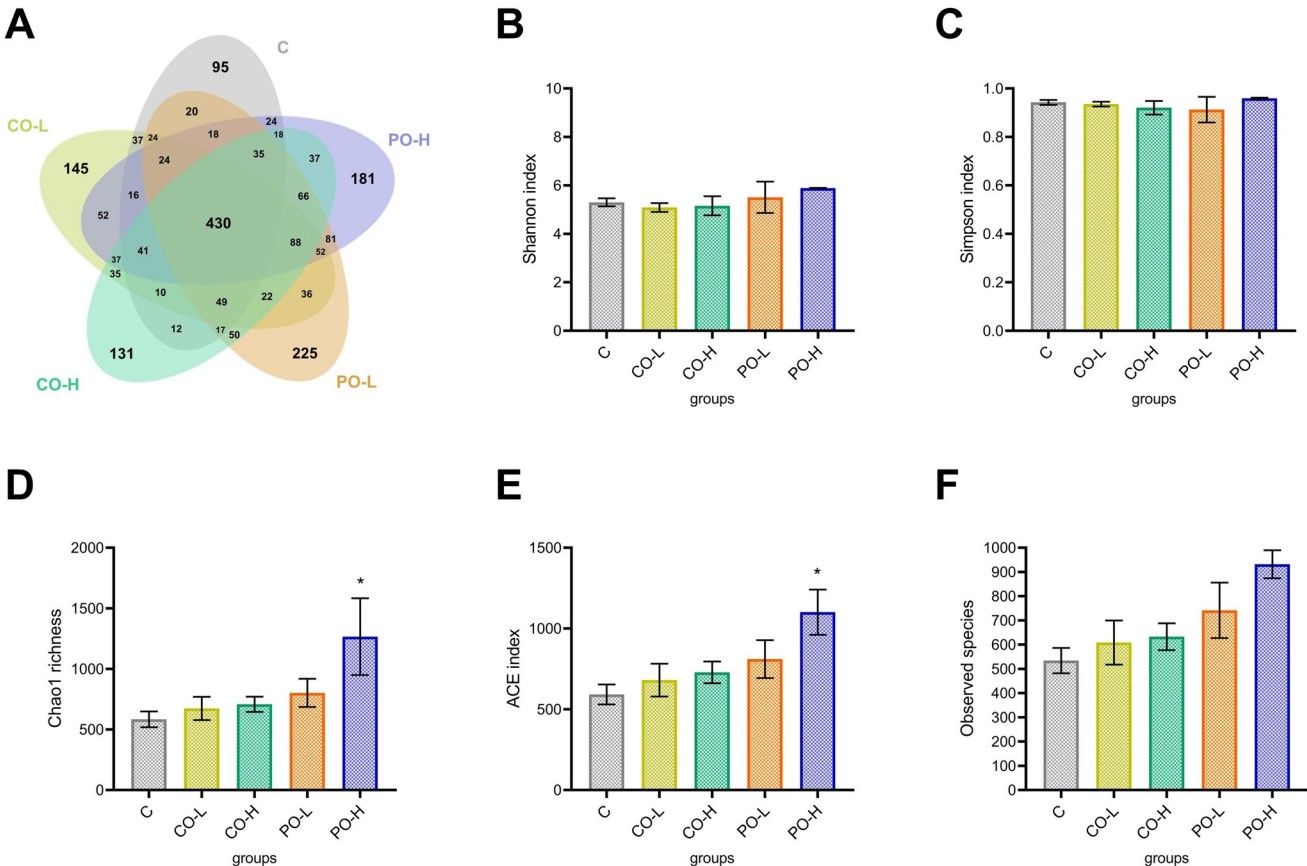

**Fig 3. Effect of CO on α-diversity of the gut microbiome community in mice.** (**A**) Venn diagram shows the number of assigned OTUs from the five experimental groups, (**B**) Shannon, (**C**) Simpson, (**D**) Chao1, (**E**) ACE, and (**F**) observed species indexes are shown. The bar graph data are expressed as mean ± SEM. * indicated the mean comparison between the treatments and control at statistical significance of $p < 0.05$.

phylum and the Firmicutes/Bacteroidetes ratio among the treatment groups, as shown in Fig 5G and 5H.

## LEfSe analysis of gut microbiota at the species level

To further access the effect of CO administration on GM in mice, the LEfSe analysis of species abundance among groups (Fig 6) found one bacterial species (*Azospirillum thiophilum*) enriched in the CO-L group, another enriched species (*Romboutsia ilealis*) in the CO-H group, two prevalent species (*Tessaracoccus rhinocerotis* and *Agrobacterium radiobacter*) in the PO-L group, and four species (*Aminobacter sp.*, *Raoultella ornithinolytica*, *Comamonas testosteroni*, *Lactobacillus reuteri*) enriched in the PO-H group, and no taxa enrichment in the C group.

## Discussion

The present study investigated the effect of CO administration on the GM community with metabolic health compared to the normal control and PO administration by using 16S rRNA metagenomic sequencing. Our results showed that CO-treated mice exhibited lower levels of food consumption than did the control group. The CO-treated group also maintained levels of caloric intake that differed from the PO-treated group. Although it seems that high-dose CO had a trend to increase the level of serum LDL, HDL, and total cholesterol, CO showed a

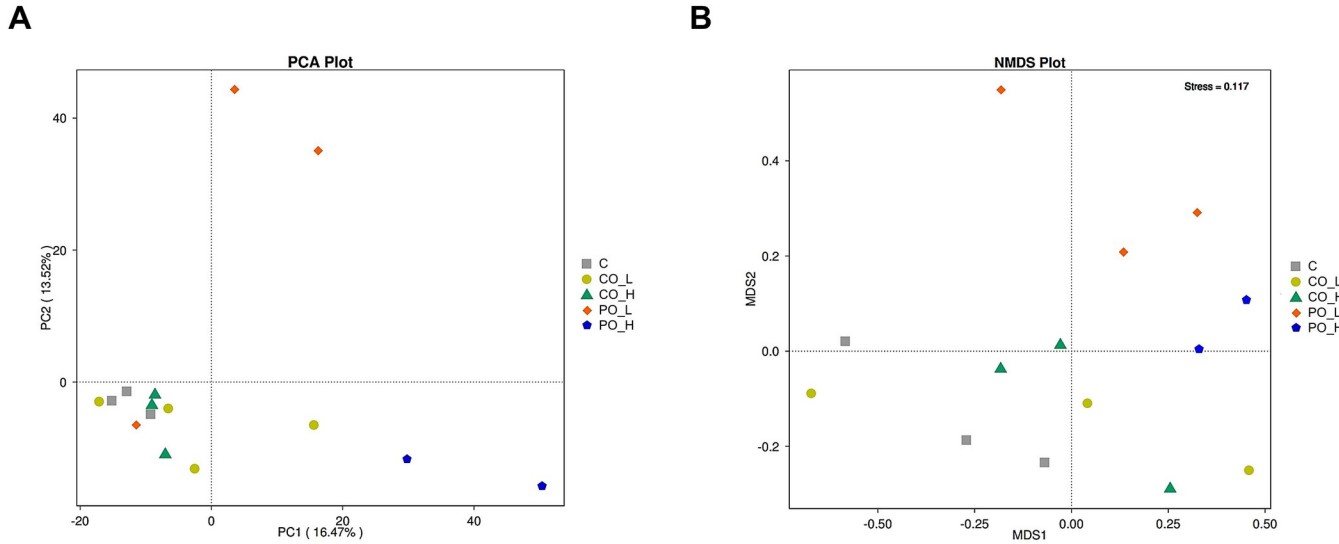

**Fig 4. Effect of CO on β-diversity of gut microbiome in mice. (A)** PCA and **(B)** NMDS of bacterial community structures are shown.

decreasing trend of serum triglyceride levels among the treatment groups. The findings also suggested that the CO groups had a similar richness and structure of the gut bacteria community, whereas the PO group altered the diversity of microbiome compositions when compared to the normal group.

Oral supplementation with a high-fat diet can play a critical role in the physiology and stability of intestine functions. GM was thought to play a vital role in high-fat diet–induced obesity and metabolic disorders [28,29]. In the present study we found that 11 wk of CO administration led to decrease average food consumption and maintain calorie intake levels when compared to the control. Our results are consistent with a previous study that reported that a high proportion of MUFAs and PUFAs in a fat diet may contribute to losing weight and reverse the state of obesity [30]. Another study also found that a high-MUFA and high-PUFA diet may decrease appetite levels by regulating the hormonal and physiological responses to dietary fatty acid composition [31]. Taking low-dose CO presented no effects on blood lipid profiles compared with those of normal mice. Nonetheless, high-dose CO showed an increase in blood cholesterol and LDL, but a decrease in triglyceride levels when compared to PO-treated mice. Recent studies indicated that a MUFA-rich diet cloud be beneficial to modulate the blood lipid profile [32,33]. Moreover, a systematic review with respect to summarize the effect of dietary fat intake on the intestinal microbiota and related metabolic health reported that a PUFA-rich diet associated with increased abundance of the phylum Tenericutes, which was associated with a lowering effect of serum triglycerides [34]. This finding suggested that CO, animal fat-source rich in MUFA and PUFA, could maintain metabolic health status. These results are consistent with our previous studies that revealed that CO exhibited a decrease in food intake, serum triglyceride levels, and the total surface area of lipid droplets in the hepatocyte [21,35]. Therefore, we suggested that the impact of CO on metabolic alterations may be associated with GM dysbiosis.

A previous study about the effect of high-fat diets on the alteration of GM reported that the high-fat diets rich in SFA, MUFA, or PUFA might exert an imbalance in the gut microbiome community [36]. The α-diversity (Chao1 and ACE indexes) showed a significant increase in the PO-H group, whereas no significant differences in the CO treatments were observed when compared to the control group. A previous study found that a high consumption of SFA might

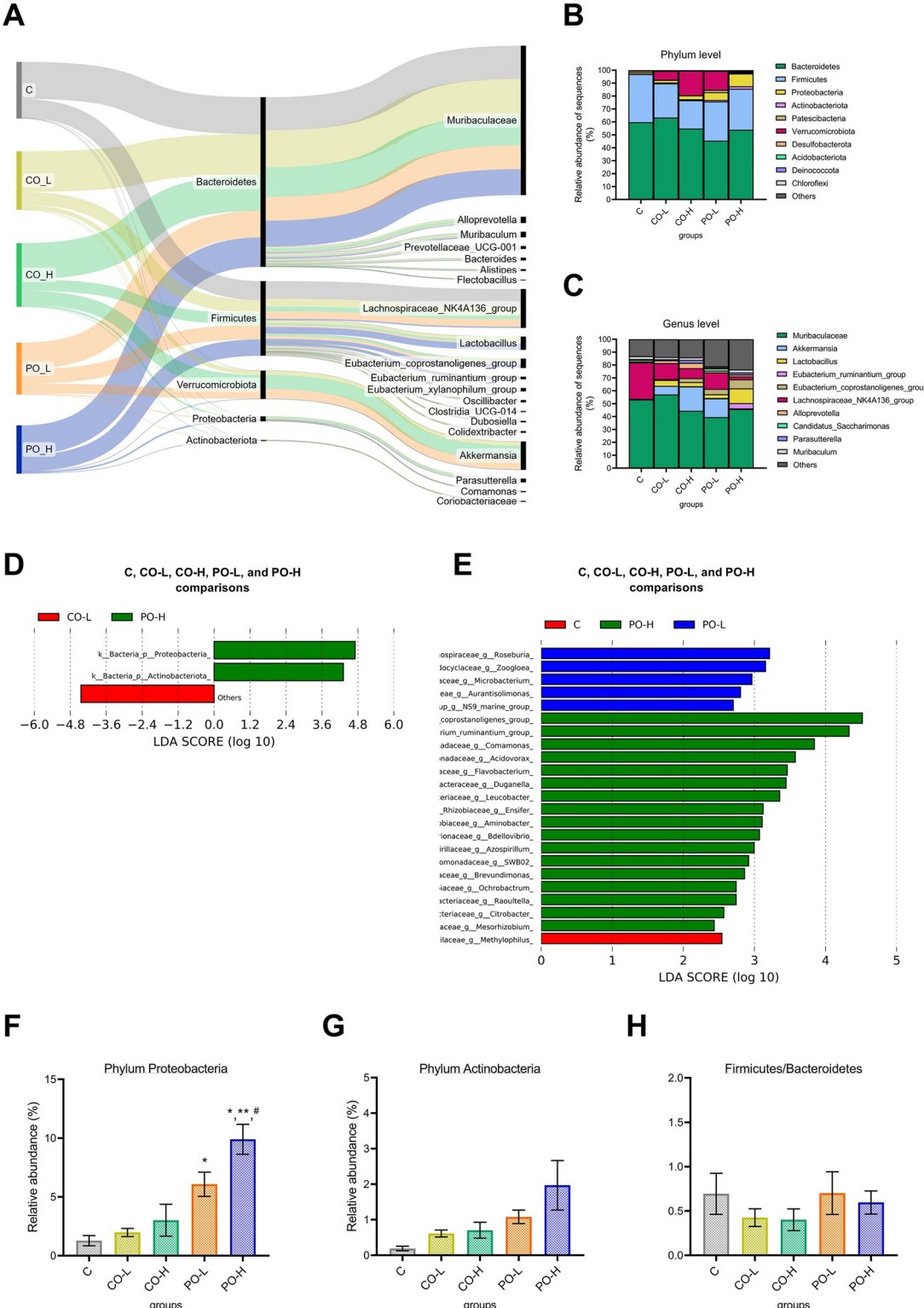

**Fig 5. Effect of CO on the relative abundance of the gut microbiome community in mice based on the 16S rRNA metagenomic sequencing data.** (**A**) Sankey diagram shows composition of the microbiome at the phylum and genus levels among the five experimental groups, (**B**) LEfSe analysis identified the most differentially abundant taxa at the phylum level of individual groups and (**C**) the most differentially abundant taxa at the genus level. Bar plots represented relative abundance of the phylum Proteobacteria (**D**), the phylum Actinobacteria (**E**), and the difference of the Firmicutes/Bacteroidetes ratio (**F**)

among these five groups. The LEfSe histogram of LDA scores show taxa with significant differences among groups. The LEfSe bar was considered at $p < 0.05$, and the threshold on the logarithmic LDA score for discriminative features was set to 2.0. The bar graph data were expressed as mean ±SEM. *, **, and # indicated the comparison within the C, CO-L, and CO-H groups at the $p < 0.05$.

unfavorably result in microbiota richness and diversity. Meanwhile, a systematic review indicated that a high consumption of MUFA might decrease total microbiota abundance and PUFA probably did not affect GM richness and diversity [34]. Interestingly, the results of the β-diversity analysis also indicated that the GM structure in the C, CO-L, and CO-H groups was more similar when compared to the PO-L and PO-H groups. These findings revealed that CO administration could accommodate and maintain the microbial community composition.

The most abundant phyla of the microbial community across five groups were Bacteroidetes and Firmicutes, followed by Verrucomicrobiota, Proteobacteria, and Actinobacteriota. The CO treatment groups showed similar levels of bacterial phyla abundance compared to the normal group. Nevertheless, an upward trend of Bacteroidetes and a downward trend of Firmicutes abundance were observed in the CO-L and the CO-H groups, especially in the CO-L group. In this regard, the CO administration also tended to decrease the Firmicutes/Bacteroidetes radio in the current study. The Bacteroidetes phylum was previously reported to be linked to metabolic disease and obesity by causing insulin resistance and altering bile acid biosynthesis and the fatty acid metabolic process [37–39]. Other studies also demonstrated that

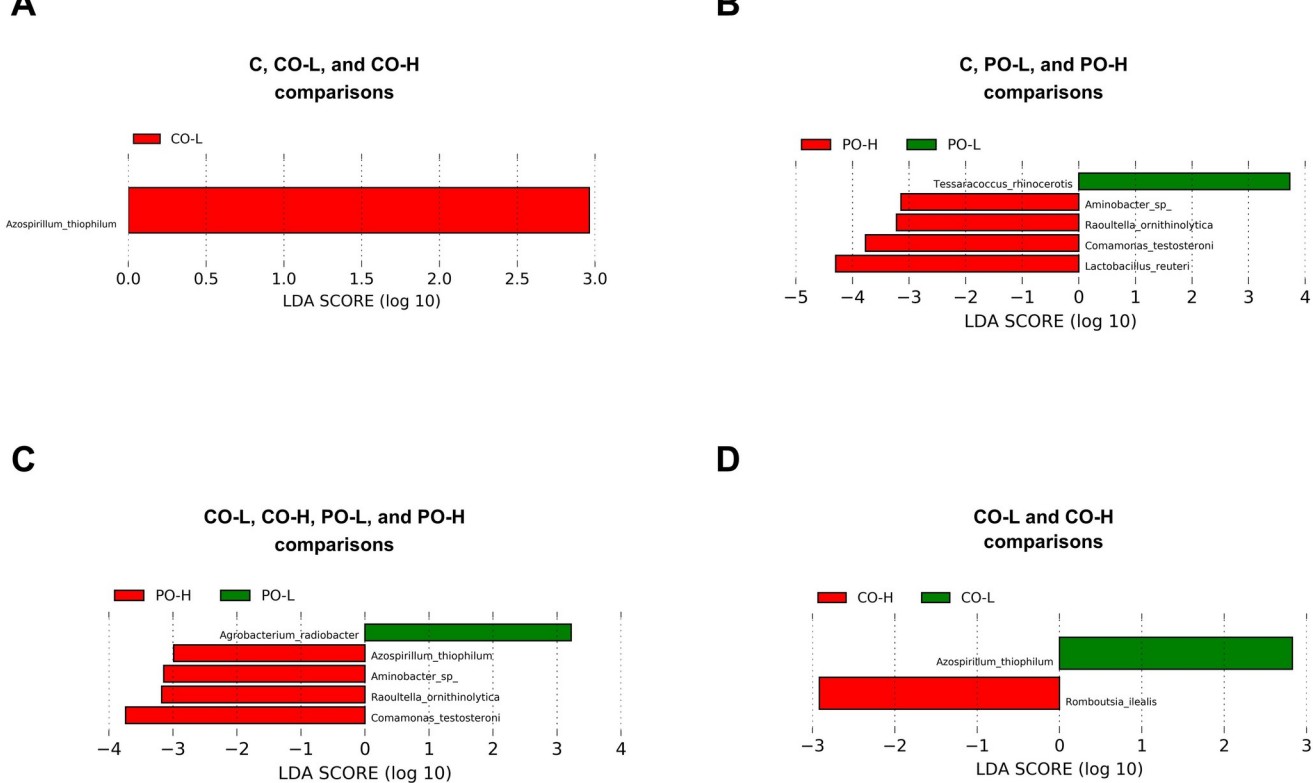

**Fig 6. LEfSe analysis of gut microbiota at the species level.** Graphs show microbiome comparison between the C, CO-L, and CO-H groups (**A**), between the C, PO-L, and PO-H groups (**B**), between the CO-L, CO-H, PO-L, and PO-H groups (**C**), and between the CO-L and CO-H groups (**D**). The LEfSe histogram of LDA scores shows taxa with significant differences among groups. The LEfSe bar at species level was considered at $p < 0.05$, and the threshold on the logarithmic LDA score for discriminative features was set to 2.0.

there was a reduction in the abundance of Bacteroidetes after high-SFA diet intake [40,41]. Furthermore, Huang et al. [42] reported that mice fed milk fat (rich in palmitic acid) exhibited an increase in Firmicutes and Proteobacteria, but a decrease in Bacteroidetes. Additionally, a previous study examined the effects of a low and high dose of PO in the gut microbiome in mice, and the results showed that a high dose of PO increased the Firmicutes/Bacteroidetes ratio, whereas a low dose of PO had the opposite effect [43]. Meanwhile, MUFA and PUFA-rich diet were reported to be associated with a reduction effect of metabolic syndromes, and obesity and improve intestinal inflammation by regulating GM [44–46]. Olive oil (rich in MUFA) supplementation was examined and showed a counteractive effect on high-fat diet–induced microbiota community changes by decreasing the Firmicutes phylum and increasing the Bacteroidetes phylum [47]. Another study on a diet rich in PUFA found not only that PUFA was correlate with increases in Bacteroidetes and decreases in Firmicutes and Proteobacteria, but it also prevented the development of obesity [48].

An elevated level of Proteobacteria abundance was found in the PO-H group, but not in the CO groups. The authors determined that a predominance of the Proteobacteria phylum from high-SFA intake was associated with the development of metabolic disorders [49,50], resulting in an increasing trend of serum triglyceride and blood glucose levels. Other research indicated that an increase in the Proteobacteria phylum was the most prominent alteration in the gut/liver axis–induced hepatic fat accumulation in rats fed a high-fructose diet [51]. The above-mentioned findings together with our results suggested that dietary CO plays a crucial role in maintaining the gut microbiome structure and proceeds to improve some metabolic symptoms in the mice model after 11 wk of experiment.

Moreover, the impact of CO consumption on bacteria alterations at the species level revealed *Azospirillum thiophilum* as the predominant species in the CO-L group. *A. thiophilum* is a well-known bacterial species inhabiting deep mineral sulfide springs with unstable physicochemical conditions [52]. Orlova et al. [53] used physiological and biochemical investigations, combined with genomic data, and demonstrated a set of genes encoding enzymes for glycolysis, the tricarboxylic acid cycle, and the electron transport chain in its genome that might favor fatty acid metabolism. Meanwhile, comparison of the dominant bacteria between the CO-L and CO-H groups indicated that *Romboutsia ilealis* was enriched in the CO-H group. *R. ilealis* is a member of the Firmicutes phylum. A recent study indicated that *R. ilealis* responded to a high-fat diet and identified it as a biomarker of gut dysbiosis [54]. This study then suggested that CO treatment could potentially promote bacterial species involved in energy source utilization and metabolic performance, but high-dose CO, a higher proportion of SFA, might result in a negative effect on the gut microbial predominant species when compared to the low-dose CO treatment.

In the current study, we also investigated whether there was a high abundance of *Roseburia* and *Zoogloea* genera in mice fed with low-dose PO. Our study is consistent with a previous study about the effect of thermally processed PO on GM in 3 months, in which their results showed that the consumption of heated PO significantly elevated the relative abundance of *Roseburia* in gut [55]. Interestingly, *Roseburia* was reported to be associated with butyrate production. Butyrate plays many important roles as a regulator of gene expression, inflammation, differentiation, and apoptosis in host cells [56,57]. However, a correlation between the *Zoogloea* genus and metabolism remains lacking. Alternatively, high-dose PO consumption stimulated an increase of the *Lactobacillus* genus, especially *Lactobacillus reuteri*. A high-SFA diet was reported to induce the expansion of *Lactobacillus* [55,58]. Zeng et al. [59] showed that an increase of many *Lactobacillus* species plays a role in development of fatty liver disease. Another study reported an association of high relative abundance of *L. reuteri* and fructose diet with adiposity and metabolic disorders [60]. Meanwhile, the administration of MUFA-

enriched oil was previously reported to induce a decrease of the *Lactobacillus* group [47]. We suggested that PO may contribute to the metabolic changes and clearly induce alteration of intestinal microbial compositions. However, other studies found that *L. reuteri* showed positive effects in health including preventing hypercholesterolemia [61] and improving diet-induced obesity [62]. With regard to this point, unsaturated fatty acid composition (MUFA and PUFA) in a fat diet may be essential for producing significant changes in the GM, which may confer metabolic health benefits. This initial finding could be further extended by increasing the sample size, prolonging the administrative duration, and testing specific fatty acids.

## Conclusions

To our knowledge, this is the first report of the effect of CO on GM related to metabolic alterations. Our findings focus on the long-term effect of different doses of CO supplementation on richness, evenness, and taxa abundances of the GM and its association with metabolism of the host. CO administration exhibited no differences in body weight content and blood glucose levels. Low-dose and high-dose CO significantly lowered food intake, but did not affect calorie intake. The CO-H fed mice showed a significant increase in serum total cholesterol and LDL levels, although they had a decreasing trend in triglyceride levels. The 16S rRNA metagenomic analysis of GM revealed that the CO-L and CO-H treatments had no influence on α-diversity, β-diversity, and relative abundance at the phylum level. In contrast, the PO-H treatment significantly increased the Choa1 and ACE indexes, distinct β-diversity, and Proteobacteria abundance. Furthermore, the abundance of *Azospirillum thiophilum* and *Romboutsia ilealis* was significantly higher in the CO-L and CO-H groups, respectively, and the species could be associated with energy metabolic activity. Different doses of CO could manage the body by having different effects on some metabolic symptoms and GM, especially at the species level. Our comprehensive study indicates that dietary CO could be considered as an alternative fat source for preserving host metabolism and intestinal microbiota. The current findings provide the evidence to use CO consumption for metabolic homeostasis with the regulating of gut flora.

## Supporting information

**S1 File.**
(XLSX)

**S2 File.**
(CSV)

## Acknowledgments

We thank International SciKU Branding (ISB), Faculty of Science, Kasetsart University for support.

## Author Contributions

**Conceptualization:** Phitsanu Tulayakul, Wirasak Fungfuang.

**Data curation:** Teerasak E-kobon.

**Formal analysis:** Kongphop Parunyakul.

**Investigation:** Teerasak E-kobon.

**Methodology:** Kongphop Parunyakul, Aphisara Chuchoiy, Sasiporn Kooltueon, Phiyaporn Puttagamnerd, Krittika Srisuksai, Pitchaya Santativongchai, Urai Pongchairerk, Phitsanu Tulayakul, Teerasak E-kobon, Wirasak Fungfuang.

**Supervision:** Wirasak Fungfuang.

**Validation:** Kongphop Parunyakul.

**Writing – original draft:** Kongphop Parunyakul.

**Writing – review & editing:** Phitsanu Tulayakul, Teerasak E-kobon, Wirasak Fungfuang.

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
