## [Decision Letter · Decision Letter 0]

17 May 2023

PONE-D-23-11866The effect of crocodile oil from Crocodylus siamensis on gut microbiome diversity with metabolic changes in micePLOS ONE

Dear Dr. Fungfuang,

Thank you for submitting your manuscript to PLOS ONE. After careful consideration, we feel that it has merit but does not fully meet PLOS ONE’s publication criteria as it currently stands. Therefore, we invite you to submit a revised version of the manuscript that addresses the points raised during the review process.

We look forward to receiving your revised manuscript.

Kind regards,

Brenda A Wilson, Ph.D.

Academic Editor

PLOS ONE

Journal Requirements:

  "This work was supported by Undergraduate Research Matching Fund (URMF), Faculty of Science, Kasetsart University and partially supported by the Faculty of Veterinary Medicine, Kasetsart University, Thailand. "

Additional Editor Comments:

Each of the reviewers noted significant issues that must be adequately addressed in a majorly revised manuscript before further consideration can be made.

Reviewers' comments:

Reviewer's Responses to Questions

**Comments to the Author**

1. Is the manuscript technically sound, and do the data support the conclusions?

Reviewer #1: No

Reviewer #2: Yes

Reviewer #3: Partly

2. Has the statistical analysis been performed appropriately and rigorously? 

Reviewer #1: Yes

Reviewer #2: Yes

Reviewer #3: No

3. Have the authors made all data underlying the findings in their manuscript fully available?

Reviewer #1: No

Reviewer #2: Yes

Reviewer #3: Yes

4. Is the manuscript presented in an intelligible fashion and written in standard English?

Reviewer #1: Yes

Reviewer #2: Yes

Reviewer #3: No

5. Review Comments to the Author

Reviewer #1: Review

Gut microbiomes are linked to metabolic health and are affected by diet. The authors investigated the effect of crocodile oil (CO) supplement on gut microbiomes and metabolic health in mice, in comparison to palm oil (PO). The authors found that CO does not affect body weight, blood glucose, or energy intake, but significantly reduced food intake. High levels of CO led to an increase in plasma total and LDL cholesterol. The authors then used 16S sequencing to characterize the gut microbiomes in these mice. The authors found that CO did not affect the diversity while the PO treatment had a more significant effect. The authors claimed that CO significantly elevated the abundance of two species which could mediate metabolic benefits, but the analysis is problematic (see below). The study is well designed with different dose groups, but the claimed conclusion is not supported by the data. For example, the overall conclusion “dietary CO could be considered as a prebiotic and alternative fat source for improving host metabolism" is not supported by the data because CO treatment had minimal effect on the gut microbiomes compared to the control group in this study.

Major comments:

1. As mentioned above, the overall conclusion “dietary CO could be considered as a prebiotic and alternative fat source for improving host metabolism" is not supported by the data because CO treatment had minimal effect on the gut microbiomes compared to the control group in this study.

2. Line 407-410 – The authors claimed “dietary CO plays a crucial role in maintaining the gut microbiome structure in the rodent model because of the high content of unsaturated fatty acids and low content of SFA when compared with the fatty acid component in PO.” The authors did not prove that it was difference in fatty acid composition that resulted in a different response from CO compared to PO.

3. The authors claimed that two species (Azospirillum thiophilum and Romboutsia ilealis) were significantly elevated in CO groups. However, the latter is identified only by comparing two treatment groups (CO-low dose vs CO-high dose) but not in the comparison with controls (6D). It would be helpful to identify species elevated in CO-high compared to controls.

4. In the conclusion section, the authors attributed the potential benefits of CO supplement to the elevated abundance of Azospirillum thiophilum and Romboutsia ilealis. The data presented in this manuscript has not established there are benefits associated with CO supplement, and the two species’ involvement in these benefits are not supported either.

Minor comments:

1. Line 102 – should be “mice” instead of “rats”

2. The source of Table 1 is not specified

3. Line 135 typo “bodyw eight”’

4. Line 232 – It was not clear what ”maintenance effect” meant

5. Line295 - should be “5D and 5E”. 5G and H were discussed but not referenced in the text

6. Line 300-305 discuss 5E, maybe better to move before the discussion of 5F

7. Line 354 – It states that CO reduces energy intake levels but it is not true. The authors wrote in other parts of the manuscript that CO does not change energy intake, so this might be a mistake.

8. Line 381, 382 – The authors stated that CO “modulate” the gut microbiome. This was not supported by the data

Reviewer #2: 

1. The introduction highlighting the essence of the crocodile oil as influential one on the diversity of gut microbiome can be cited with string references

1. In methodology use of reference can be more respect to each methodology ;standard diet of mice with respect to other lipid substances interference can be highlighted to ensure the methodology to be reproducible

2. The interesting finding of the author was to new gut bacteria Roseburia and Zoogloea and Azospirillum thiophilum and Romboutsia ilealis with their significant impact on gut microbiome diversity analysis well expressed

3. Although the concept of work is bit different in present scenario, still if more current references can be included in discussion and methodology it fetch more relevance with the data presented.

4. The research tools, various statistical programme packages are in its proper way utilized

5. The interpretation of the results quite rightly befits ; discussions can be more from recent bibliography expected

6. The essence of crocodile oil in gut microbiome study indeed a recent boom in research and in that aspect the works befits the gap of the research.

I finally recommend the article with minor revisions suggested to accept for further publication criteria

Reviewer #3: 

This study addresses the effects of the oil extracted from the fatty tissues of the Siamese crocodile Crocodylus siamensis (referred to as crocodile oil) on the composition of the gut microbiota and the metabolism of laboratory mice. For this purpose, five groups of mice received sterile water (control), 1 or 3% (v/w) crocodile oil, or 1 or 3% (v/w) palm oil for 11 weeks. The results seemed to indicate that the administration of crocodile did not lead to impressive differences in either the composition of the gut flora or the carbohydrate and lipid metabolism of the mice. Nevertheless, the authors concluded that ‘Different doses of crocodile oil had different therapeutic effects on some metabolic symptoms and gut microbiota’, and suggested that ‘The CO may be an alternative fat source and benefit the host metabolism by maintaining the GM structure’.

General comments

Both the conclusion(s) and the implication(s) of the study do not seem sufficiently supported by the results obtained. Therefore, the conclusions from this study seem farfetched and must be downplayed.

The use of the English language is must be improved. The authors are recommended to have a native English speaker review the manuscript before resubmitting it.

Specific comments

Title

The authors may consider to change to: ‘Effects of the oil from the fatty tissues of Crocodylus siamensis on gut microbiome diversity and metabolism of laboratory mice

Abstract

Better structured; must immediately be understandable as a stand-alone text.

Line 27. More specifics about the crocodile oil

Line 45. What was ‘the C group’?

Lines 47-48. The conclusion ‘Different doses of CO had different therapeutic effects on some metabolic symptoms and GM’ is exaggerated and thus not correct, and must be downplayed.

Lines 48-50. The implication of the conclusion ‘The CO may be an alternative fat source and benefit the host metabolism by maintaining the GM structure’ must also be downplayed

Introduction

Line 54: Before using an abbreviation, When mentioned for the first time, the entire expression must be given with the abbreviation between brackets, after which the abbreviation can be used throughout the rest of the text. Thus: ‘The gut microbiota (GM) …….’

Lines 102-105. ‘We focused on the relationship between the GM response to dietary fat and metabolic changes to define the therapeutic properties that shape and maintain the healthy GM compositions in the host’. The authors must more clearly indicate the potential impact of the use of crocodile oil on the one hand, and ‘the properties that shape and maintain the healthy GM compositions in the host’.

Materials and Methods

Lines 126-127. ‘The fatty acid composition of fats used is listed in Table 1’. How was the fatty acid composition of the fats determined?

Lines 147-149. ‘The serum lipid profiles included triglycerides, cholesterol, HDL, and LDL and were determined using a Hitachi 7080 analyzer (Hitachi, Tokyo, Japan)’. Which method was used to determine serum lipid profiles? Lines 182-183. ‘Quantitative experimental data are expressed as mean ± SEM. Statistical analysis was performed by one-way ANOVA followed by a Tukey’s post hoc test ……’. Why not means ± SDs? Too few repeats? And if so, was ANOVA justified?

Results

The authors may consider to present their data in tables instead of figures.

Discussion and Conclusions

This section is needlessly lengthy with respect to the results obtained and must be written in a more concise, matter-of-fact fashion.

And as mentioned before, the conclusions from the study seem farfetched and must be downplayed.

References

The references must consequently be written according to the format of the journal.

6. PLOS authors have the option to publish the peer review history of their article (what does this mean?). If published, this will include your full peer review and any attached files.

Reviewer #1: No

Reviewer #2: No

Reviewer #3: No

---

## [Author Response · Author response to Decision Letter 0]

15 Jun 2023

Response to reviewers

Reviewer #1: Review

Gut microbiomes are linked to metabolic health and are affected by diet. The authors investigated the effect of crocodile oil (CO) supplement on gut microbiomes and metabolic health in mice, in comparison to palm oil (PO). The authors found that CO does not affect body weight, blood glucose, or energy intake, but significantly reduced food intake. High

levels of CO led to an increase in plasma total and LDL cholesterol. The authors then used 16S sequencing to characterize the gut microbiomes in these mice. The authors found that CO did not affect the diversity while the PO treatment had a more significant effect. The authors claimed that CO significantly elevated the abundance of two species which could mediate metabolic benefits, but the analysis is problematic (see below). The study is well designed with different dose groups, but the claimed conclusion is not supported by the data. For example, the overall conclusion "dietary CO could be considered as a prebiotic and alternative fat source for improving host metabolism" is not supported by the data because CO treatment had minimal effect on the gut microbiomes compared to the control group in this study.

Major comments:

1. As mentioned above, the overall conclusion "dietary CO could be considered as a prebiotic and alternative fat source for improving host metabolism" is not supported by the data because CO treatment had minimal effect on the gut microbiomes compared to the control group in this study.

Response: Thank you for your concern and nice recommendation. We have checked, downplayed, and revised the overall conclusion to conform to our data as follows: “Our comprehensive study indicates that dietary CO could be considered as an alternative fat source for preserving host metabolism and intestinal microbiota. The current findings provide the evidence to use CO consumption for metabolic homeostasis with the regulating of gut flora.”, please see pages 21-22, lines 479-483.

2. Line 407-410 - The authors claimed "dietary CO plays a crucial role in maintaining the gut microbiome structure in the rodent model because of the high content of unsaturated fatty acids and low content of SFA when compared with the fatty acid component in PO." The authors did not prove that it was difference in fatty acid composition that resulted in a different response from CO compared to PO.

Response: Thank you for your concern and nice recommendation. We have checked, and downplayed this paragraph of the discussion, please see pages 18-19, lines 389-412. 

3. The authors claimed that two species (Azospirillum thiophilum and Romboutsia ilealis) were significantly elevated in CO groups. However, the latter is identified only by comparing two treatment groups (CO-low dose vs CO-high dose) but not in the comparison with controls (6D). It would be helpful to identify species elevated in CO-high compared to controls.

Response: Thank you for your concern and nice recommendation. We have checked already and we think that the comparison among 3 groups (C, CO-L, and CO-H) was identified as shown in Figure 6A, and the results showed that Azospirillum thiophilum enriched in the CO-L group but there had no prevalent species from the C and CO-H. Additionally, the analysis of species abundance among groups was adequate on this point, please considerer this matter. 

4. In the conclusion section, the authors attributed the potential benefits of CO supplement to the elevated abundance of Azospirillum thiophilum and Romboutsia ilealis. The data presented in this manuscript has not established there are benefits associated with CO supplement, and the two species' involvement in these benefits are not supported either.

Response: Thank you for your concern and nice recommendation. We have checked and modified the conclusion section to improve the message of our finding as follows: “Furthermore, the abundance of Azospirillum thiophilum and Romboutsia ilealis was significantly higher in the CO-L and CO-H groups, respectively, and the species could be associated with energy metabolic activity. Different doses of CO could manage the body by having different effects on some metabolic symptoms and GM, especially at the species level. Our comprehensive study indicates that dietary CO could be considered as an alternative fat source for preserving host metabolism and intestinal microbiota. The current findings provide the evidence to use CO consumption for metabolic homeostasis with the regulating of gut flora.”, please see page 21, lines 475-483.

Minor comments:

1. Line 102 - should be "mice" instead of "rats"

Response: Thank you for your nice comment. We have checked and revised the word, please see page 5; line 101.

2. The source of Table 1 is not specified

Response: Thank you for your nice comment. The data shown in Table 1 is adapted from previous studies. We have checked and added the source of Table 1 in the table legends, please see page 7; line 136. 

3. Line 135 typo "bodyw eight"'

Response: Thank you for your notice. We have checked and revised this typo already, please see page 7; line 138. 

4. Line 232 - It was not clear what "maintenance effect" meant

Response: Thank you for your concern. We have checked, downplayed, and revised as follows: “favorable effect”, please see page 12; line 235.

5. Line295 - should be "5D and 5E". 5G and H were discussed but not referenced in the text

Response: Thank you for your concern. We have checked and revised as follows: “The linear discriminant analysis (LDA) effect size (LEfSe) analysis at the phylum levels (Fig 5D) showed differences in the abundances of Proteobacteria and Actinobacteriota among the groups. Meanwhile, LEfSe analysis at the genus levels (Fig 5E) showed higher abundances of Methylophilus in the C group, Roseburia and Zoogloea in the PO-L group, and the Eubacterium coprostanoligenes group, the Eubacterium ruminantium group, and Comamonas in the PO-H group, whereas the predominant genus was not observed in the CO-L and CO-H groups.”, please see page 14; lines 297-303.

6. Line 300-305 discuss 5E, maybe better to move before the discussion of 5F

Response: Thank you for your nice recommendation. We have checked and revised as follows: “Meanwhile, LEfSe analysis at the genus levels (Fig 5E) showed higher abundances of Methylophilus in the C group, Roseburia and Zoogloea in the PO-L group, and the Eubacterium coprostanoligenes group, the Eubacterium ruminantium group, and Comamonas in the PO-H group, whereas the predominant genus was not observed in the CO-L and CO-H groups. The Proteobacteria phylum had significantly highest abundant in the PO-H group followed by the PO-L, whereas the C, CO-L and CO-H treatment had lower abundance levels (Fig 5F).”, please see page 14, lines 299-306.

7. Line 354 - It states that CO reduces energy intake levels but it is not true. The authors wrote in other parts of the manuscript that CO does not change energy intake, so this might be a mistake.

Response: Thank you for your nice recommendation and reminding the authors. We have checked and revised the mistake as follows: “In the present study we found that 11 wk of CO administration led to decrease average food consumption and maintain calorie intake levels when compared to the control”. Please see pages 16-17, lines 355-357. 

8. Line 381, 382 - The authors stated that CO "modulate" the gut microbiome. This was not supported by the data

Response: Thank you for your concern and nice recommendation. We have checked, downplayed, and revised as follows: “These findings revealed that CO administration could accommodate and maintain the microbial community composition.” Please see page 18; lines 386-388.

Reviewer #2:

1. The introduction highlighting the essence of the crocodile oil as influential one on the diversity of gut microbiome can be cited with string references

Response: Thank you for your concern and nice recommendation. Due to our study is the first report on the effect of crocodile oil on gut microbiota, there had not enough recent studies. However, we have cited the research about the effect of other fat sources on the gut microbiome already, please see revised Introduction and please considerer this matter. 

1. In methodology use of reference can be more respect to each methodology; standard diet of mice with respect to other lipid substances interference can be highlighted to ensure the methodology to be reproducible

Response: Thank you for your concern and nice recommendation. Each treatment used in our study (sterile water, crocodile oil, and palm oil) was daily fed as a supplement, and the animals were fed a normal standard diet with the same ingredient. Thus, the authors can ensure that our methodology could be reproducible.

2. The interesting finding of the author was to new gut bacteria Roseburia and Zoogloea and Azospirillum thiophilum and Romboutsia ilealis with their significant impact on gut microbiome diversity analysis well expressed

Response: Thank you for your kindness and nice comment.

3. Although the concept of work is bit different in present scenario, still if more current references can be included in discussion and methodology it fetch more relevance with the data presented.

Response: Thank you for your concern and nice recommendation. We have checked and given more current references and data sources in the discussion and methodology section already, please see revised Discussion and Material and Method. 

4. The research tools, various statistical programme packages are in its proper way utilized

Response: Thank you for your kind consideration.

5. The interpretation of the results quite rightly befits; discussions can be more from recent bibliography expected

Response: Thank you for your kind consideration. We have checked and given more suitable detail already, please see revised Result and Discussion 

6. The essence of crocodile oil in gut microbiome study indeed a recent boom in research and in that aspect the works befits the gap of the research.

Response: Thank you for your kind consideration and nice comment.

I finally recommend the article with minor revisions suggested to accept for further publication criteria

Response: Thank you so much for your kind consideration.

Reviewer #3:

This study addresses the effects of the oil extracted from the fatty tissues of the Siamese crocodile Crocodylus siamensis (referred to as crocodile oil) on the composition of the gut microbiota and the metabolism of laboratory mice. For this purpose, five groups of mice received sterile water (control), 1 or 3% (v/w) crocodile oil, or 1 or 3% (v/w) palm oil for 11 weeks. The results seemed to indicate that the administration of crocodile did not lead to impressive differences in either the composition of the gut flora or the carbohydrate and lipid metabolism of the mice. Nevertheless, the authors concluded that 'Different doses of crocodile oil had different therapeutic effects on some metabolic symptoms and gut microbiota', and suggested that 'The CO may be an alternative fat source and benefit the host metabolism by maintaining the GM structure'.

General comments

Both the conclusion(s) and the implication(s) of the study do not seem sufficiently supported by the results obtained. Therefore, the conclusions from this study seem farfetched and must be downplayed. The use of the English language is must be improved. The authors are recommended to have a native English speaker review the manuscript before resubmitting it.

Specific comments

Title

The authors may consider to change to: 'Effects of the oil from the fatty tissues of Crocodylus siamensis on gut microbiome diversity and metabolism of laboratory mice

Response: Thank you for your nice recommendation. We have checked and revised as follows: “Effect of the oil from the fatty tissues of Crocodylus siamensis on gut microbiome diversity and metabolism in mice”, please see page 1; lines 1-3.

Abstract

Better structured; must immediately be understandable as a stand-alone text.

Line 27. More specifics about the crocodile oil

Response: Thank you for your recommendation. We have modified the whole abstract as a stand-alone text already and added more detail of specifics about the CO as follows: “Crocodile oil (CO) was extracted from the fat tissues of Crocodylus siamensis. CO, rich in monounsaturated- and polyunsaturated fatty acids, has been reported to improve inflammation, toxification, and energy metabolism.”, please see page 2; lines 27-30.

Line 45. What was 'the C group'?

Response: Thank you for your reminder. The C group is referred to the Control group, we have given more detail as follows: “orally administrated with sterile water (control [C])”, please see page 2, lines 32-33.

Lines 47-48. The conclusion 'Different doses of CO had different therapeutic effects on some metabolic symptoms and GM' is exaggerated and thus not correct, and must be downplayed.

Lines 48-50. The implication of the conclusion 'The CO may be an alternative fat source and benefit the host metabolism by maintaining the GM structure' must also be downplayed

Response: Thank you for your nice comment. We have checked and downplayed as follows: “Furthermore, the abundance of Azospirillum thiophilum and Romboutsia ilealis was significantly higher in the CO-L and CO-H groups which could be associated with energy metabolic activity. Thus, CO may be an alternative fat source to improve the host metabolism by maintaining the gut flora.”, page 2; lines 46-49.

Introduction

Line 54: Before using an abbreviation, when mentioned for the first time, the entire expression must be given with the abbreviation between brackets, after which the abbreviation can be used throughout the rest of the text. Thus: 'The gut microbiota (GM) …….'

Response: Thank you for your concern and nice recommendation. We have checked and revised, please see page 3; line 53.

Lines 102-105. 'We focused on the relationship between the GM response to dietary fat and metabolic changes to define the therapeutic properties that shape and maintain the healthy GM compositions in the host'. The authors must more clearly indicate the potential impact of the use of crocodile oil on the one hand, and 'the properties that shape and maintain the healthy GM compositions in the host'.

Response: Thank you for your comment and nice recommendation. We have checked and revised as follow: “We discuss how CO interacts with the GM systems and the relationship between gut microbes and host metabolism. Our research might provide new insights into the potential impact of the use of CO on GM homeostasis, which is associated with host metabolism and intestinal health.” Please see page 5; lines 101-105.

Materials and Methods

Lines 126-127. 'The fatty acid composition of fats used is listed in Table 1'. How was the fatty acid composition of the fats determined?

Response: Thank you for your concern and recommendation. We have checked and added more detail as follows: “The total PUFAs and MUFAs contents were higher and SFAs contents were lower in CO than that of PO.”, please see page 6; lines 126-129. 

Lines 147-149. 'The serum lipid profiles included triglycerides, cholesterol, HDL, and LDL and were determined using a Hitachi 7080 analyzer (Hitachi, Tokyo, Japan)'. Which method was used to determine serum lipid profiles?

Response: Thank you for your concern. We have checked and given more detail as follow: “The serum lipid profiles included triglycerides, cholesterol, HDL, and LDL and were enzymatically determined on a HITACHI-7080 automatic biochemical analyzer (Hitachi, Tokyo, Japan).”, please see page 7; lines 150-152.

Lines 182-183. 'Quantitative experimental data are expressed as mean ± SEM. Statistical analysis was performed by one-way ANOVA followed by a Tukey's post hoc test ……'. Why not means ± SDs? Too few repeats? And if so, was ANOVA justified?

Response: Thank you for your concern. Our study would like to emphasize small differences in our data because the distribution of our data is not normal, and we need to cover it up by showing the data as Mean ± SEM instead. On the order hand, we perform a one-way ANOVA with small sample sizes. Some problems with small sample sizes are associated with low statistical power and an inflated false discovery rate, and the majority of statistical programs for analysis could handle the problems and run ANOVA as usual. Thus, we think that the ANOVA could work with small samples from each group in this case. 

Results

The authors may consider to present their data in tables instead of figures.

Response: Thank you for your concern and recommendation. We have checked and considered to present Figure 1B-G in the table form, please see Table 2; page 11.

Discussion and Conclusions

This section is needlessly lengthy with respect to the results obtained and must be written in a more concise, matter-of-fact fashion. And as mentioned before, the conclusions from the study seem farfetched and must be downplayed.

Response: Thank you for your concern and nice recommendation. We have checked, revised, and downplayed the whole discussion and conclusions, please see the revised discussion and conclusions. 

References

The references must consequently be written according to the format of the journal.

Response: Thank you for your concern. We have checked and adjusted the references into the PlosOne format, pleases see the revised references.

---

## [Decision Letter · Decision Letter 1]

29 Jun 2023

PONE-D-23-11866R1Effect of the oil from the fatty tissues of Crocodylus siamensis on gut microbiome diversity and metabolism in micePLOS ONE

Dear Dr. Fungfuang,

Thank you for submitting your manuscript to PLOS ONE. After careful consideration, we feel that it has merit but does not fully meet PLOS ONE’s publication criteria as it currently stands. Therefore, we invite you to submit a revised version of the manuscript that addresses the points raised during the review process.

Both reviewers felt that the manuscript was substantially improved scientifically and there are only a few remaining issues to address. However, both noted that there is still a need for improvement regarding English grammar and syntax. Please revise accordingly.

We look forward to receiving your revised manuscript.

Kind regards,

Brenda A Wilson, Ph.D.

Academic Editor

PLOS ONE

Journal Requirements:

Additional Editor Comments:

The manuscript still requires some English editing for grammar and syntax.

Reviewers' comments:

Reviewer's Responses to Questions

**Comments to the Author**

1. If the authors have adequately addressed your comments raised in a previous round of review and you feel that this manuscript is now acceptable for publication, you may indicate that here to bypass the “Comments to the Author” section, enter your conflict of interest statement in the “Confidential to Editor” section, and submit your "Accept" recommendation.

Reviewer #1: (No Response)

Reviewer #3: All comments have been addressed

2. Is the manuscript technically sound, and do the data support the conclusions?

Reviewer #1: Partly

Reviewer #3: Yes

3. Has the statistical analysis been performed appropriately and rigorously? 

Reviewer #1: Yes

Reviewer #3: Yes

4. Have the authors made all data underlying the findings in their manuscript fully available?

Reviewer #1: No

Reviewer #3: Yes

5. Is the manuscript presented in an intelligible fashion and written in standard English?

Reviewer #1: Yes

Reviewer #3: No

6. Review Comments to the Author

Reviewer #1: For comment #1: The authors have corrected the conclusion in the the discussion but not in the abstract. The authors should modify the abstract to be consistent. Specifically, the conclusion "Thus, CO may be an alternative fat source to improve the host metabolism by maintaining the gut flora" is not supported as improvement of the host metabolism and is not demonstrated.

Reviewer #3: The authors have clearly done their best to take into account the referees' comments when revising the manuscipt. As a result, the scientific quality of the manuscript has considerably improved. This particulalrly holds true for the conclusions of the experimental findings, which have been presented in a less exaggerated, more realist fashion.

However, it is clear that the authors are not native English speakers. Not surprisingly, there are many spelling and syntactic errors which must be corrected.

A few examples in only the Abstract are:

- line 28: 'fat tissues'; must be changed to 'fatty tissues'

- lines 29-30: 'has been reported to improve inflammation, toxification, and energy metabolism' must be changed

to, for instance, 'has been reported to reduce inflammation, counter toxification, and improve energy metabolism'

- line 31: 'on gut microbiota (GM) with metabolic changes in mice' must be changed to, for instance, 'on gut

microbiota (GM) in labotratory mice as well as the accompanying metabolic changes in the animals'

- lines 33-34: 'administrated with sterile water (control [C]), 1 and 3% (v/w) CO (CO-low [CO-L] and CO-high

[CO-H]), or 1 and 3% (v/w) palm oil (PO-low and PO-high) for 11 weeks' must be changed to 'administrated either

sterile water (control [C]); 1 or 3% (v/w) CO (CO-low [CO-L] and CO-high [CO-H]), respectively); or 1 or 3%

(v/w) palm oil (PO-low and PO-high, respectyively) for 11 weeks'

- line 36: 'Colon samples' must be changed to 'Samples from colon tissue'

- line 37: clarify GM analyses

- lines 38-40: 'CO-L and CO-H groups showed a significant reduction in food intake, but they had no effect on calorie

intake when compared to the C group' must be changed to, for instance, 'Food intake by the mice in the CO-L and

CO-H groups was statistically significantly less when compared to that by the the animals in the C group. However,

neither CO treatment had a statistically significant effect on calorie intake when compared to the controls'.

- Etc.

7. PLOS authors have the option to publish the peer review history of their article (what does this mean?). If published, this will include your full peer review and any attached files.

Reviewer #1: No

Reviewer #3: No

---

## [Author Response · Author response to Decision Letter 1]

8 Jul 2023

Response to reviewers

Reviewer #1: 

Reviewer #1: For comment #1: The authors have corrected the conclusion in the the discussion but not in the abstract.The authors should modify the abstract to be consistent. Specifically, the conclusion "Thus, CO may be an alternative fat source to improve the host metabolism by maintaining the gut flora" is not supported as improvement of the host metabolism and is not demonstrated.

Response: Thank you for your concern and nice recommendation. We have checked, downplayed, and revised the conclusion in the abstract already as follows: “Thus, CO may be an alternative fat source for preserving host metabolism and gut flora.”, please see pages 3, lines 51-52.

Reviewer #3:

The authors have clearly done their best to take into account the referees' comments when revising the manuscipt. As a result, the scientific quality of the manuscript has considerably improved. This particulalrly holds true for the conclusions of the experimental findings, which have been presented in a less exaggerated, more realist fashion. 

Response: Thank you for your concern and nice recommendation. The scientific quality and writing in our results, discussion, and conclusion have been improved in a more realist fashion. 

A few examples in only the Abstract are: 

- line 28: 'fat tissues'; must be changed to 'fatty tissues'

- lines 29-30: 'has been reported to improve inflammation, toxification, and energy metabolism' must be changed to, for instance, 'has been reported to reduce inflammation, counter toxification, and improve energy metabolism'

- line 31: 'on gut microbiota (GM) with metabolic changes in mice' must be changed to, for instance, on gut microbiota (GM) in labotratory mice as well as the accompanying metabolic changes in the animals'

- lines 33-34: 'administrated with sterile water (control [C]), 1 and 3% (v/w) CO (CO-low [CO-L] and CO-high [CO-H]), or 1 and 3% (v/w) palm oil (PO-low and PO-high) for 11 weeks' must be changed to 'administrated either sterile water (control [C]); 1 or 3% (v/w) CO (CO-low [CO-L] and CO-high [CO-H]), respectively); or 1 or 3% (v/w) palm oil (PO-low and PO-high, respectyively) for 11 weeks'

- line 36: 'Colon samples' must be changed to 'Samples from colon tissue'

- line 37: clarify GM analyses

- lines 38-40: 'CO-L and CO-H groups showed a significant reduction in food intake, but they had no effect on calorie intake when compared to the C group' must be changed to, for instance, Food intake by the mice in the CO-L and CO-H groups was statistically significantly less when compared to that by the the animals in the C group. However, neither CO treatment had a statistically significant effect on calorie intake when compared to the controls'.

Response: Thank you for your concern and nice recommendation. We have checked and revised already, please see the revised Abstract (pages 2-3; lines 26-52).

---

## [Editor Report · Decision Letter 2]

11 Jul 2023

Effect of the oil from the fatty tissues of Crocodylus siamensis on gut microbiome diversity and metabolism in mice

PONE-D-23-11866R2

Dear Dr. Fungfuang,

We’re pleased to inform you that your manuscript has been judged scientifically suitable for publication and will be formally accepted for publication once it meets all outstanding technical requirements.

Kind regards,

Brenda A Wilson, Ph.D.

Academic Editor

PLOS ONE

Additional Editor Comments (optional):

Reviewer concerns appear to be adequately addressed.

---

## [Editor Report · Acceptance letter]

21 Jul 2023

PONE-D-23-11866R2 

Effect of the oil from the fatty tissues of *Crocodylus siamensis* on gut microbiome diversity and metabolism in mice 

Dear Dr. Fungfuang:

I'm pleased to inform you that your manuscript has been deemed suitable for publication in PLOS ONE. Congratulations! Your manuscript is now with our production department. 

Kind regards, 

on behalf of

Dr. Brenda A Wilson 

Academic Editor

PLOS ONE